# Neurorehabilitation in Multiple Sclerosis—A Review of Present Approaches and Future Considerations

**DOI:** 10.3390/jcm11237003

**Published:** 2022-11-27

**Authors:** Carmen Adella Sîrbu, Dana-Claudia Thompson, Florentina Cristina Plesa, Titus Mihai Vasile, Dragoș Cătălin Jianu, Marian Mitrica, Daniela Anghel, Constantin Stefani

**Affiliations:** 1Department of Neurology, ‘Dr. Carol Davila’ Central Military Emergency University Hospital, 010242 Bucharest, Romania; 2Department of Rehabilitation Medicine, Elias Emergency University Hospital, 011461 Bucharest, Romania; 3Alessandrescu-Rusescu National Institute for Mother and Child Health, Fetal Medicine Excellence Research Center, 020395 Bucharest, Romania; 4Department of Preclinical Disciplines, Titu Maiorescu University, 031593 Bucharest, Romania; 5Clinical Neurosciences Department, University of Medicine and Pharmacy “Carol Davila”, 050474 Bucharest, Romania; 6Centre for Cognitive Research in Neuropsychiatric Pathology (Neuropsy-Cog), Department of Neurology, Faculty of Medicine, “Victor Babeș” University of Medicine and Pharmacy, 300041 Timișoara, Romania; 7Department of Medico-Surgical and Prophylactic Disciplines, Titu Maiorescu University, 031593 Bucharest, Romania; 8Department of Family Medicine and Clinical Base, ‘Dr. Carol Davila’ Central Military Emergency University Hospital, 010242 Bucharest, Romania; 9Department No. 5, University of Medicine and Pharmacy “Carol Davila”, 050474 Bucharest, Romania

**Keywords:** multiple sclerosis, rehabilitation, gait, balance, fatigue, spasticity, dysphagia, overactive bladder, neurorehabilitation

## Abstract

Multiple sclerosis is an increasingly prevalent disease, representing the leading cause of non-traumatic neurological disease in Europe and North America. The most common symptoms include gait deficits, balance and coordination impairments, fatigue, spasticity, dysphagia and an overactive bladder. Neurorehabilitation therapeutic approaches aim to alleviate symptoms and improve the quality of life through promoting positive immunological transformations and neuroplasticity. The purpose of this study is to evaluate the current treatments for the most debilitating symptoms in multiple sclerosis, identify areas for future improvement, and provide a reference guide for practitioners in the field. It analyzes the most cited procedures currently in use for the management of a number of symptoms affecting the majority of patients with multiple sclerosis, from different training routines to cognitive rehabilitation and therapies using physical agents, such as electrostimulation, hydrotherapy, cryotherapy and electromagnetic fields. Furthermore, it investigates the quality of evidence for the aforementioned therapies and the different tests applied in practice to assess their utility. Lastly, the study looks at potential future candidates for the treatment and evaluation of patients with multiple sclerosis and the supposed benefits they could bring in clinical settings.

## 1. Introduction

Multiple sclerosis (MS) is an immunologically driven pathology affecting the central nervous system, characterized by chronic inflammation and progressive demyelinating lesions, with an unidentified etiology [1]. MS is currently affecting 2.8 million people worldwide, while in North America and Europe it is the leading cause of chronic non-traumatic neurological disease in young adults [2]. The prevalence is higher in women (69%) than in men (31%), and the number of children below the age of 18 reported to suffer from MS is continuously increasing [2].

The symptoms caused by multiple sclerosis cover a wide spectrum of neurological impairments, due to the nature of the lesions, which can be located in various areas of the central nervous system. However, the most common of them include diplopia (double vision), loss of sight in one or more areas of the visual field, nystagmus, dysphagia (difficulty swallowing solids, liquids or both), dysphonia, cognitive function impairments, alterations in all types of sensitive perception, fatigue, gait and balance disorders, ataxia, spasticity, and bowel and bladder disorders [3,4]. In addition to the above-mentioned impairments, walking is also gradually affected, specifically the speed and distance covered without the occurrence of fatigue, leading to an increased dependence in the activities of daily living (ADL) and a decreased quality of life (QoL) [5,6]. To assess the level of disability and therapeutic approaches in clinical settings for patients suffering from multiple sclerosis, Dr. John Kurtzke developed in the 1950s the Kurtzke Disability Status Scale (DSS), which has since been modified several times and led to the currently used 10-point Expanded Disability Status Scale (EDSS) [7]. The EDSS measures gait and eight additional functional systems (FS): pyramidal (motor function), cerebellar, brainstem, sensory, bowel and bladder, visual, cerebral or mental and other. The scores start at 0, which translates into a normal neurological exam; 1–3 corresponds to a mild disability, without signs of affected ambulation; 3.5–5.5 represents a moderate disability, with patients starting to display ambulation restrictions; a score of 6–6.5 requires walking aids; 7–8 refers to the need to use a wheelchair; in the 8.5 to 9.5 range, the patient is generally restricted to bed and a score of 10 corresponds to death due to MS [8].

One of the treatments used for multiple sclerosis consists of drug therapies that have the capacity to positively influence the rate of relapses, the progression of lesions on MRI (magnetic resonance imaging) scans, as well as the overall evolution of the disease [9]. However, while the majority of these medicines are able to improve certain parameters, such as ambulation capability, fatigue, and spasticity to various degrees, studies show they have little effect on pre-existing neurological deficits [10,11].

Neurorehabilitation has only recently been considered a treatment option in the context of multiple sclerosis, and is generally being used either as a supportive therapy for the control of symptoms or as a preventive approach for the consequences related to a sedentary lifestyle [12,13,14]. Studies show that MS patients that are included in rehabilitation programs improved their quality of life and are more independent in their activities of daily living [15,16]. Furthermore, several symptoms associated with MS are beneficially impacted through exercise, such as cardiovascular capacity [17], neuromuscular function [18], ambulation [19], depression [20], and cognitive performance [21]. Recently, neuroimaging techniques have also revealed the positive effects of neurorehabilitation on the anatomy and physiology of the brain, together with markers of inflammation [22,23,24,25].

Neurorehabilitation is a therapeutic option for all multiple sclerosis patients that is constantly adapting and improving together with the advancement of technology, making it an increasingly affordable and easy-to-self-administer approach [26]. With the advent of the COVID-19 pandemic, professionals have made more use of novel techniques to allow patients to participate remotely in programs through telerehabilitation [27]. Moreover, various types of passive exercise technologies, which can target specific brain areas involved in MS, are being implemented in everyday practice, such as noninvasive brain stimulation (NIBS) and robot-assisted therapy devices [28]. More specifically, repetitive transcranial magnetic stimulation (rTMS) as a form of NIBS has proven its utility in the treatment of cognitive deficits, spasticity and fatigue in the context of multiple sclerosis [29,30].

## 2. The Effect of Neurorehabilitation on the Neurobiological Particularities of Multiple Sclerosis Patients

The central nervous system (CNS) possesses a characteristic called neuroplasticity, which can be defined by its ability to adapt and remodel as a result of environmental pressure, exerted upon itself by disease or injury [31]. This adaptive response triggers alterations in the neuroglia (changes of number and size), the grey matter (the building of new synapses, dendritic branching modifications and axonal sprouting) and the white matter (myelin production, fiber density alterations) [31]. Neuroplasticity can be noticed subsequent to neurological lesions such as stroke, and has also been documented in patients with MS, potentially compensating for the damage caused by the demyelination processes [11]. Research involving functional magnetic resonance imaging (fMRI) highlights the ability of MS patients’ brains to continuously reorganize, but in an apparently limited manner in severe cases of MS, possibly due to the extension of the underlying lesions [31]. A study conducted by Bonzano et al. demonstrated that the neuroplasticity in MS could be preserved through neurorehabilitation programs designed to extend over predetermined periods of time [32]. Moreover, other research involving fMRI scans showed that in comparison to the control groups, non-disabled MS patients used more energy when asked to perform simple tasks and presented more activated areas in the brain [33]. Neuroplasticity influences a variety of functions, such as memory, cognition, and motor function [34,35,36,37].

The literature regarding the fMRI changes in the brains of MS patients also analyzed the impacts of cognitive rehabilitation [38]. Following cognitive rehabilitation programs, resting-state MRI neuroimaging revealed an improvement in the patterns of brain synchronization and cognitive performance, involving areas in the right frontal middle orbital gyrus and the visual medial resting state network (RSN), from the cerebellum crus 1 region, which corresponded to the clinically observed improved performances [38]. Furthermore, other studies achieved similar results when using the classic block-design fMRI technique [39,40,41,42], thus underlining the importance of the cerebellum in performing executive tasks and in the process of cognition.

Nevertheless, changes occurring in the architecture of the brain through neuroplasticity can also be detrimental to the individual, by sustaining or contributing to the preexisting disability [43,44]. To exemplify this, a number of studies have suggested that brain plasticity is preserved regardless of the severity of the cerebral pathology, as long as rehabilitation focuses on repeating a task for a sufficient amount of time [43,44,45], while others report a decreased or potentially absent adaptive capacity in patients suffering from a primary progressive form of MS, as opposed to those with relapsing–remitting MS [46]. Furthermore, research shows that the newly formed network connections in the brains of MS patients have a higher complexity level than the previous architecture of the healthy brain, unlike other conditions, such as stroke, where brain tissue restoration follows its original network patterns [47,48,49,50,51,52]. In addition, the neuroplasticity of patients with MS could decline after two years of an initial increase, thus leading to a progression of the disease and disability [53,54].

Besides the changes in neuroplasticity determined by neurorehabilitation programs, the exercises involved can also induce peripheral immunomodulatory responses [55,56]. Research focusing on experimental autoimmune encephalomyelitis (EAE) in mice found that endurance and resistance training protocols could enhance the immunosuppressive functions by elevating the markers of regulatory T lymphocytes (Treg), therefore leading to an improvement in neurological disability [57], and this was further confirmed through passive immunization [58,59]. Consequential benefits were also observed regarding cytokine levels, infiltrating immune cells, astrogliosis, and microgliosis [55,56]. Recently, a novel regulatory connection between the peripheral immune system and the hypothalamus was discovered in EAE mice through environmental enrichment. The research highlights the immunomodulatory activity determined by the effect of the brain-derived neurotrophic factor (BDNF), produced in the hypothalamus, upon the glucocorticoid receptor in thymocytes [60]. Likewise, demyelinating models in mice, using cuprizone (CPZ) and lysolecithin (LCT), showed the potential of voluntary exercise to determine myelination and direct anti-inflammatory effects, through reduced microgliosis, astrogliosis and the loss of myelin, and enhanced myelin production capacity and the proliferation of oligodendrocyte precursor cells (OPCs), respectively [61,62].

Lastly, neurorehabilitation programs may have beneficial effects on the gut microbiota of patients with a long history of MS, which could improve the level of inflammation related to the disease [63]. Past studies observed various levels of dysbiosis in patients suffering from multiple sclerosis when compared to healthy individuals, with a severe depletion in short-chain fatty acids (SCFAs)-producing bacteria from the Lachnospiraceae family [64,65,66]. One of the SCFAs that is of particular importance in the context of multiple sclerosis is butyrate, a bacterial metabolite that is involved in preserving the integrity of the intestinal barrier, and also in the process of Treg differentiation [67,68,69,70,71]. Furthermore, higher levels of bacteria involved in pro-inflammatory responses were detected in the case of MS patients, with gut microbiota enriched in species such as Prevotella and Collinsella [63,72,73]. The reassessment of the individuals involved in the study following a complex rehabilitation routine revealed an improved clinical status, namely, improved fatigue and gait, in tight correlation with reduced inflammatory markers, particularly the pro-inflammatory cytokine IL-17 and a more balanced gut microbiota [63].

## 3. Present Therapeutic Approaches

Multiple sclerosis causes a wide range of symptoms determined by various lesional patterns in the central nervous system that lead to a degree of handicap. While the relapsing–remitting form most commonly displays visual and sensory deficiencies (46% and 41% respectively), primary–progressive MS typically displays gait impairments (88%) and various degrees of paresis (38%) [74]. Symptoms may have a variable influence on the quality of life of each patient, with fatigue being the most commonly reported disturbance of everyday life activities [75]. The median survival time is around 40 years from the moment it was diagnosed; therefore, many MS patients report a variable degree of disability throughout the course of the disease. About 29% of patients require a wheelchair and 50% are using walking aids 15 years following the diagnosis [76]. Furthermore, the median time of retirement is 11.1 years from diagnosis, which is significantly lower than in the average population [77]. Treatments that aim to improve the symptoms of multiple sclerosis are therefore essential, and must take a multidisciplinary approach between a number of medical specialties, requiring medication, neurorehabilitation, and psychological therapy.

### 3.1. Disease-Modifying Therapies and Symptomatic Medication in Multiple Sclerosis

Patients diagnosed with multiple sclerosis are required to follow a strict drug therapy course for the rest of their lives. The gold standard treatment for acute relapses is represented by intravenous steroids administered in high doses to diminish the inflammatory damage and accelerate the recovery process [78]. Pro-inflammatory metabolite clearance through plasmapheresis could also be considered in cases that do not show an improvement after steroids [79]. Chronic medical treatment includes various classes of drugs, amongst which immunomodulatory medicines represent the first-line agents [80]. Beta interferons (IFNβ) were the first disease-modifying therapies to be approved in 1993, offering clinicians a valuable tool to reduce the number of relapses and to postpone the onset of disability in relapsing–remitting MS patients [81]. Other drugs utilized for the treatment of relapsing–remitting MS include injectable therapies such as glatiramer acetate and oral therapies such as fingolimod, dimethyl fumarate, diroximel fumarate, teriflunomide, Siponimod, and cladribine [82]. The only available treatment for primary–progressive MS is ocrelizumab [82]. Other lines of treatment aim at improving the debilitating symptoms associated with the progression of the disease. Therefore, drug therapies with gabapentin, tizanidine or baclofen have been proven effective in reducing spasticity [83,84]; botulinum toxin injections or oral oxybutynin could improve overactive bladder symptoms [85,86]; fatigue management could be achieved using modafinil or amantadine [87], while lamotrigine, gabapentin and carbamazepine could alleviate sudden pain attacks [88,89]. However, while effective in managing the disease, the above-mentioned treatments do not stop the progression of multiple sclerosis.

### 3.2. Physical Rehabilitation Strategies

The focus of rehabilitation is to help patients with multiple sclerosis acquire the best possible recovery, allowing them to reduce their physical and mental impairments and offering them the possibility to remain completely or partially integrated in society. For instance, the current guidelines of the National Health Service (NHS) in the UK offer patients suffering from MS a two-week rehabilitation program, consisting of a personalized number of daily sessions of the following therapies: physiotherapy, occupational therapy, speech and language therapy, diet and nutrition advice and neuropsychology [90]. Further, this paper will discuss the current approaches to neurorehabilitation for the most common symptoms of multiple sclerosis. It is worth mentioning that patients with multiple sclerosis might present a variety of the following symptoms in different degrees of severity. Various techniques can be used for treating multiple symptoms. The goal of rehabilitation is to improve the quality of life of MS patients, by targeting the most debilitating symptoms for each individual, without overexerting them. Thus, when putting together a neurorehabilitation routine for a specific MS patient, the practitioner should consider using techniques that are both efficient and target more than one symptom that particular patient is presenting. The symptoms and overlapping techniques used for their treatment are presented in Figure 1.

#### 3.2.1. Gait Management

Walking impairment is one of the most frequent and debilitating symptoms experienced by people diagnosed with multiple sclerosis, with up to 93% of them experiencing a variable degree of gait limitation 10 years after diagnosis [91,92]. Objectively, walking disabilities can be measured in a clinical setting using well-established tests, such as the 2-Minute Walk Test (2MWT), the 6-Minute Walk Test (6MWT) [93], the Timed 25-Foot Walk test (T25FW) [94] and the 12-Item Multiple Sclerosis Walking Scale (MSWS-12) [95]. These tests are useful tools for assessing the activity of the disease, and offer the possibility of evaluating treatment efficacy.

The majority of patients with multiple sclerosis present muscle weakness, more frequently in the trunk and lower limbs. This is considered one of the most important contributing factors that determine gait impairment [96,97,98]. Strength training is therefore crucial, and should be performed at least twice every week in the process of rehabilitation for people with MS [99]. Moreover, studies have shown that this type of physical activity is beneficial for maintaining neuroplasticity through the activation of motor units and firing rate synchronization [100]. Strength training exercises can be performed using a variety of techniques, machines and weight levels, and can target different muscle groups, with all of these different approaches offering similar outcomes [101]. When it comes to weights, some clinicians prefer using solely the body weight of the patient performing the exercises [102,103], while others choose weight machines, resistance bands [104] or cuff weights [105]. The weight machines most commonly utilized are the traditional ones [106,107,108], or may include isokinetic dynamometers [109]. Finally, the typically targeted muscle groups include the ones involved in knee extension [105,106,107,108], knee flexion [106,110,111], hip extension, flexion [104,106,108] and abduction [103,104], and ankle flexion [109] and extension [110], with most programs consisting of a combination of the abovementioned.

A complementary type of exercise for people with gait deficiencies is endurance training—for example, walking and cycling, which aim to improve aerobic capacity and allow MS patients to walk increasingly longer distances [112]. However, due to the increased risk of falling, body weight-supported treadmill training (BWSTT) is preferred—an exercise that can be useful in the early initiation phase [113]. A novel and more efficient way of performing BWSTT is through robotic-assisted gait training (RAGT), which is more stable, provides a reduced workload for the physiotherapist and is more physiological and reproducible [114]. While most studies focus on progressive resistance training, there is also an alternative approach to gait training. One particular exercise that can be replicated on body weight-supported devices is speed-intensive gait training, involving alternating short intervals of walking at faster speeds with longer periods of walking at a normal pace [115]. This can enhance endurance, speed and other measurable parameters, both in the healthy population [116,117] and in patients suffering from neurological impairments [118,119].

Ankle–foot orthoses (AFOs) represent a frequently recommended solution for gait, balance and strength improvement [120]. Previous studies suggest that their utility is more pronounced in people with higher levels of disability [121]. Recently, clinicians have also started recommending textured insoles for similar purposes, which have shown promising results after repeated plantar stimulation for more than two weeks [122,123].

#### 3.2.2. Balance and Coordination Management

Balance and coordination impairments are some of the most common issues reported by patients suffering from multiple sclerosis. Exercises that focus on balance improvement should aim at preventing falls, and enhancing walking stability and posture control, while those targeted at improving coordination should reduce energy requirements and increase the continuity of movement. Frenkel exercises are commonly used for this purpose. They consist of slow repetitions of each stage of movement that gradually increase in complexity and require high levels of concentration. For instance, the action of sitting up is split into three phases—withdrawing the feet, bending the trunk forward, straightening the legs while getting up [124]. In order to improve the accuracy of exercises for each individual case, patients that are capable of standing without support can be required to perform the exercises on a stabilometric platform [125]. Balance and coordination training can be complemented by proprioception exercises, which further decrease the risk of injury in patients affected by balance impairments [126]. An alternative to the abovementioned is hippotherapy, which uses the natural movement of a horse to improve balance and gait in people suffering from various neurological conditions [127].

A certain degree of variation is required during the rehabilitation process of people with MS, due to the lengthy periods of time involved, which could determine a lower level of motivation and compliance to treatment. Therefore, the Bobath concept could be used as an option to improve the outcomes of these patients [128]. The Bobath concept, also known as neurodevelopmental treatment, is a problem-solving approach, which assumes dysfunctional postural reflexes needed for movement coordination and equilibrium are the essential cause of motor deficits in people with central nervous system lesions [124]. It focuses on inhibiting pathological tonic reflexes in order to achieve appropriate active motion and muscle tension. The approach is also more convenient for people with higher EDSS scores, thanks to the fact that it can be applied in a multitude of positions, including supine and prone positions.

Proprioceptive neuromuscular facilitation (PNF) is a rehabilitation technique that can also be used to improve balance and coordination, together with mobility and spasticity [129,130]. It enhances the muscle function through stimulation of the proprioceptive organs present in tendons and muscles, thus improving postural reflexes and increasing balance, strength and flexibility [131,132]. The method has been extensively studied and proven efficient in patients with post-stroke impairments, and requires further assessment in patients with MS, as it could provide a valuable addition to their treatment.

#### 3.2.3. Fatigue Management

Another frequent symptom reported by MS patients is fatigue, which is encountered in 75–95% of cases [133,134,135] and is considered a key factor affecting the quality of life in these people [136,137]. Fatigue is defined as a perceived reduction in physical and mental energy that hinders everyday activities [135]. Physical exercise, especially aerobic training, can improve both primary and secondary fatigue in MS patients, through direct changes in the central nervous system and inflammation reduction, but also by improving depression symptoms and quality of sleep [138]. Physical exercises for fatigue management should be adapted to each individual, taking into consideration the patient’s degree of disability [139]. These include strength exercises, aerobic training (walking, running, swimming, cycling), neuromotor exercises (dancing, tai chi, yoga, pilates) and breathing exercises [138].

In addition to physical training, physiotherapy procedures should be used to enhance the effects of exercise. Approaches using high temperatures should be avoided, considering the negative impact of heat on nerve conductivity and fatigue [140]. Cryotherapy has proven its potential benefits on fatigue management in several instances, either through whole-body cryotherapy, which involves short sessions of whole-body exposure to ultra-low temperatures (−110 °C) [141], or by using a cooling garment [142]. However, patients with certain conditions, such as hypertension, cardiovascular diseases, a history of blood clotting or thyroid sufferance, should not be exposed to cryotherapy [141].

Another procedure for MS patients with fatigue is pulsed electromagnetic field therapy (PEMF). One of the advantages of this technique is that it offers the option of using it at home, through a small, portable device [143]. One of the routines studied involved 8 min sessions, two times a day for 12 weeks, which resulted in significant improvements in the level of perceived fatigue [144].

Training programs can also be enhanced by functional electrical stimulation (FES) in patients with MS [145,146]. One study analyzed the effects of muscular electrical stimulation through FES during cycling. The researchers observed an amelioration of pain, fatigue and cognitive impairment after 24 weeks of training [145]. In another paper, FES was applied on the quadricep muscles during training, showing beneficial effects on fatigue levels after 8 weeks [146].

#### 3.2.4. Spasticity Management

A symptom that is particularly debilitating in MS is spasticity. This can involve all four limbs, with an increased predilection towards the lower extremities, and it is measured by the Modified Ashworth Scale (MAS) that ranges from 0 (no increase in tone) to 4 (flexion and extension are limited in the examined part). In moderation, spasticity can exert beneficial effects on blood circulation and can counter muscle atrophy. However, beyond a certain level, it leads to joint malformations, contractures and pressure ulcers. Spasticity is present in 40–60% of MS patients [124].

The management of spasticity requires a spectrum of therapies including medication (baclofen administered through intrathecal or oral route), transcranial magnetic stimulation, botulinum toxin injections and physiotherapy [147,148,149]. Of the abovementioned, physiotherapy is the most utilized treatment applied to patients living with spasticity [150]. The approaches taken by rehabilitation programs to treat this impairment range from physical training to vibration therapy (focal muscle vibration or whole-body vibration), hydrotherapy, electrical stimulation, radial shock wave therapy, electromagnetic fields, cryotherapy, and therapeutic standing on an Oswestry standing frame [149]. Their aim is to maintain neuroplasticity, prevent contracture and preserve the length of muscles [151].

Rehabilitation plans should not employ intense physical efforts, which can aggravate spasticity, and instead should prioritize the use of physical agents ahead of exercise training, for enhancing the efficiency of the latter [124]. Cryotherapy is one of the procedures that can be used prior to exercise initiation, and is utilized either systemically through whole-body cryotherapy and ice baths, or locally through cryo cuffs, cooling garments and ice massage [152,153]. Its purpose is to induce local anesthesia and reduce the reaction to active stretching.

Physical training to alleviate spasticity should be introduced gradually, starting with lighter exercises and avoiding intense stretching, and should concentrate on improving the range of motion of the ankle dorsiflexion, decreasing the muscle tone in the calf, and enhancing the strength of the antigravity muscles [154,155]. Literature reviews have found significant improvements in spasticity and MAS in patients engaged in BWSTT and RAGT [156,157], and in those performing outpatient exercises, such as walking, endurance, aquatic, active and passive stretching exercises [149].

Electrotherapy is another physiotherapeutic method for alleviating spasticity. It can be applied in the form of transcutaneous electrical nerve stimulation (TENS), functional electrical stimulation (FES), neuromuscular electrical stimulation (NMES), and Hufschmidt electrical stimulation. TENS is typically used for the treatment of pain; however, it can provide an alternative treatment for spasticity, if used prior to physical training [158,159]. The method uses electrodes placed on dermatomes or along the nerves, delivering frequencies that range between 1 and 100 Hz. The frequency can be adapted to every individual level of feeling, yet most studies involving spastic paresis used frequencies of 99–100 Hz [159]. The intensity ranged between 15 and 50 mA and the impulse duration between 0.06 and 0.2 ms [159]. In contrast, FES uses rectangular pulse currents with lower frequencies of 20–50 Hz and pulse widths of 0.1–0.2 ms, applied on paretic muscles [124]. NMES is a more efficient procedure that is not influenced by motor neuron damage. It uses electrical impulses that are stronger and wider than the ones used in TENS. The functional parameters are usually established through electromyography (EMG) investigations [160].

Electromagnetic fields represent a further type of therapeutic intervention that improves spasticity in MS patients. According to the literature, they can be delivered through either transcranial magnetic stimulation [161,162], pulsed electromagnetic field therapy (PEMF) [143,163] or repetitive peripheral magnetic nerve stimulation (RPMS) [164,165]. The advantages of using peripheral electromagnetic fields in the treatment of spasticity reside in the fact that they induce significantly less pain than other types of electrotherapy, such as NMES [166], they are permeable through human tissues, and they do not produce heat [124]. The frequencies cited by various studies are placed within the 1–150 Hz range [167], with impulses having trapezoidal, sinusoidal, rectangular, or triangular shapes [124].

Water can provide a good environment, and presents a multitude of benefits, for MS patients. Hydrotherapy decreases the activity of gamma neurons, and limits the afferent impulses, leading to a relaxing and analgesic effect, and finally to a reduction in spasticity. The preferred temperature is between 34 and 36 °C, and hot baths are not permitted in order to avoid the occurrence of the Uhthoff effect [168]. Water also acts as a supportive medium for physical exercises, without presenting the risk of falling, therefore increasing mobility in patients with multiple sclerosis. In addition, hydrotherapy improves fatigue and depression symptoms [169].

#### 3.2.5. Dysphagia Management

Dysphagia is a symptom that occurs in around 43% of MS patients [170], and is the result of a number of factors such as cognitive impairment, cranial nerves paresis and accumulated lesions in the brainstem, cerebellum and the corticobulbar tracts [171]. If left untreated, it can seriously impact the quality of life and lead to life-threatening consequences such as malnutrition, dehydration and aspiration pneumonia [172]. Since it can be associated with speech disorders, the utilized therapies often focus on treating the two symptoms associatively, through physiotherapy, occupational therapy and speech–language therapy (SLT) [173].

SLT is an essential tool in the rehabilitation of MS patients suffering from dysphagia, which aims at re-teaching swallowing in order to prevent food aspiration. The therapy includes exercises that strengthen the muscle structures involved in swallowing, the stimulation of the deglutition and cough reflexes (for defense purposes), posture training for the head and trunk, and the establishment of compensatory actions for more natural swallowing [174]. For example, in a case study described by Farazi et al., which rendered positive results, SLT procedures were provided two to three times every day, for two weeks. One of the techniques involved was compensatory swallow therapy through progressively increasing the quantity and consistency of the intake. Further, oral motor exercises including passive and active movements, as well as massage, were included. Chin-down posture training and the Mandelson method were also part of the program [173].

Physiotherapy can provide useful tools for managing dysphagia through physical exercises, botulinum toxin injections and electrotherapy [174]. Weight management and appetite stimulation are some of the beneficial effects of increased physical activity. Botulinum toxin treatment is administered in the cryopharingeus muscle under general or local anesthesia for upper esophageal sphincter dysfunctions [175]. It can be injected either through esophagoscopy [176] or percutaneously through electromyographic guidance [177]. Lastly, transcranial direct current stimulation is another viable therapeutic approach for MS patients with swallowing difficulties. Recent studies have demonstrated a mild and transient improvement in deglutition scores when the current was applied over the right swallowing motor cortex for five consecutive days [178].

#### 3.2.6. Overactive Bladder Management

Between 63% and 68% of patients with multiple sclerosis develop neurogenic bladder dysfunction during the course of the disease [179]. Therapeutic strategies that target it include pelvic floor muscle training (PFMT), also known as Kegel exercises, bladder training, drug therapies, electrostimulation therapy and botulinum toxin injections [74].

Pelvic floor muscle training is aimed at increasing the resting tension of the pelvic diaphragm, through a series of repetitions of tensing and relaxing the groups of muscles in the region [180]. Strengthening these muscles allows them to contract for prolonged periods, therefore enhancing the control over the mechanism of urination. Other muscle groups, such as the adductor, the gluteal and the transversus abdominis muscles, should be strengthened together with PFMT, as they present a reduced activation in patients with urinary incontinence [181]. Bladder training programs that aim to increase the capacity of the bladder through behavioral changes [182] can also be associated, and if necessary, weight loss should be considered in order to reduce physical stress over the bladder [183].

Electrostimulation therapy is another alternative for the management of an overactive bladder. Electrostimulation can be vaginal or anal, and it is directed at stimulating the pudendal nerves while inhibiting hyperreflexia [74]. Moreover, botulinum toxin injected in the detrusor muscle via cystoscopy also has the ability to suppress hyperreflexia. Studies have shown that doses of 100–150 units were effective for 3–6 months [180].

### 3.3. Cognitive Rehabilitation

Besides the physical disabilities caused by the brain lesions in multiple sclerosis, cognitive impairments are also associated in 34–65% of cases, depending on the duration of the disease and the age at onset [184]. These include memory deficits, diminished speed of processing and attention, and other symptoms linked with a decreased cognitive reserve. The brain structures that show a particularly affected connectivity are the cortical prefrontal lobe and the amygdala in the limbic system, responsible for emotional control [185]. Together with cognitive deficits, this leads to the further development of depression and other emotional disorders [186].

In treating these afflictions, neurocognitive rehabilitation needs to be associated with psychotherapy for the optimal therapeutic outcome. Disease severity or duration seem to be less related to the development of emotional disorders than inadequate acceptance and coping mechanisms, therefore downgrading medication to a second-line treatment option [187]. Cognitive reappraisal, coping improvement and stress management are all effective strategies for treating depression and improving the quality of life of these patients [188,189]. Cognitive behavioral therapy (CBT) is an example of psychotherapy that enables cortical prefrontal structures to mitigate negative emotional responses by adjusting the patients’ perception and stress levels in situations outside their control [190]. Neurocognitive rehabilitation is a useful tool in ameliorating cognitive impairments in MS patients [191]. It is targeted at the brain’s neuroplasticity through retraining certain functions, such as memory, attention, learning and executive functions [186]. Thanks to the development of knowledge regarding neuroplasticity, new and improved methods are now available for the treatment and diagnosis of cognitive deficits, including aerobic exercises and transcranial direct current stimulation (tDCS) [192].

## 4. Subjective and Objective Measures of Improvement after Neurorehabilitation

The internet medical databases contain numerous studies on a variety of rehabilitation procedures applied to MS patients, with the aim of alleviating their diverse symptoms. However, while some of them provide moderate- to high-quality evidence for their benefits, others do not offer such reliability. This may be due to the difficulty in designing double-blinded studies in the area of rehabilitation, or to the subjective nature of some of the tests used for assessment. Further, this paper will analyze some of the evidence provided for the neurorehabilitation procedures involved in treating MS, together with the reliability of the tests and scales that are applied.

Quality of life (QoL) is a complex assessment test that encompasses a wide spectrum of domains covering the elaborate definition of health provided by the World Health Organization [193]. In multiple sclerosis, QoL is impacted by a multitude of factors, such as impairments affecting everyday activities, level of dependency on caregivers, employment status, social support, or mental health [194,195,196,197,198]. In this context, QoL assessment provides useful information concerning the progression of the disease or the impact of therapy [199,200]. Various rehabilitation interventions have been evaluated using QoL, mostly in the areas of cognitive rehabilitation, psychotherapy, and the treatment of fatigue [201,202,203,204], all of which have shown success in improving QoL scores. Furthermore, other studies that focused on improving social support also enhanced the QoL [205,206].

The activities of daily living (ADL) scale is another useful measure for multiple sclerosis patients, which focuses more on the physical impairment aspect, but it can also provide insights related to cognition [207,208]. Rehabilitation programs aimed at improving mobility, fatigue and cognitive deficits have been assessed using the ADL scale. In 2019, a comprehensive Cochrane meta-analysis was published, with the purpose of evaluating the quality of evidence (according to the GRADE framework) provided by previous review papers that analyzed the impacts of various rehabilitation techniques on ADL [16]. Three randomized control studies from one review, comprising a total of 217 participants, provided a moderate quality of evidence that multidisciplinary inpatient rehabilitation is beneficial for improving mobility, functional independence (ADL) and locomotion (for patients using a wheelchair) [209,210,211]. However, the authors note that these studies provided strong evidence for ADL (and the similar Barthel index) improvement due to rehabilitation, but the quality of evidence was downgraded because different outcome measures were used. A moderate quality of evidence was also provided regarding the efficacy of inpatient or outpatient rehabilitation on improving bladder impairment, and the ability of exercise training to enhance mobility, muscle strength and effort tolerance [16]. Evidence for balance improvement using whole-body vibration techniques and for the short-term benefits of telerehabilitation on functional activities was graded as low quality, due to the increased risk of bias and the use of different outcome measures in the analyzed studies [16].

Gait rehabilitation is among the main goals in the treatment of multiple sclerosis. The deficits are caused by a range of factors including sensory disturbances, cerebellar impairments, spasticity, and muscle weakness that lead to a markedly decreased quality of life [212]. In clinical settings, gait assessment is commonly performed using timed walking tests (2MWT, 6MWT, T25FW) or standardized scales (EDSS). However, these measures do not offer great reliability, due to the limits of timed walking tests used for evaluating gait quality [213,214,215] or to the low sensitivity of EDSS to short-term changes [216]. A more accurate method to assess the effect of neurorehabilitation on gait deficits is represented by novel technologies in the form of wearable sensors [217]. These devices are able to track the subtle changes in gait kinematics while performing a surface electromyography (sEMG) that detects spasticity through muscle activation patterns [218,219]. In a study performed by Huang et al., a 4-week multidisciplinary gait rehabilitation program was assessed using wearable technology. They reported significant progress in gait speed, kinematics, spasticity and balance, in alignment with improved results in the standard clinical tests [220]. Moreover, wearable accelerometers could also be a useful tool in monitoring gait kinematics in MS patients, even in non-clinical settings. Researchers suggest that the future benefits of accelerometers reside in their potential to become a biomarker for disease severity and progression [221].

Various neurorehabilitation strategies have been evaluated for balance and coordination improvement in MS patients. Currently, there are a number of tests that are used to track different static and dynamic parameters that influence these two functions. The trunk impairment scale (TIS), Berg balance scale (BBS), international cooperative ataxia rating scale (ICARS) and nine-hole peg test (NHPT) are among the tests performed in clinical settings. TIS is a reliable test applied to patients with multiple sclerosis that uses a selection of movements to evaluate three parameters: coordination, static sitting balance and dynamic sitting balance [222]. The Berg balance scale represents another widely utilized test comprising 14 items that provide information on the patient’s balance abilities and the changes induced in them by rehabilitation programs [223]. Ataxia is an MS symptom that can be assessed using ICARS, a valid and reliable scale containing four subscales targeting posture and gait disorders, limb ataxia, dysarthria and oculomotor impairments [224]. Lastly, NHPT is a test that evaluates manual dexterity, affecting up to 75% of MS patients [225], and it is currently considered the gold standard in its field [226]. Previous studies demonstrated significant improvements in all the above-mentioned tests after both exercises based on the Bobath method and traditional rehabilitation routines [227]. In order to improve the objectiveness of measurement, some authors have developed innovative solutions, such as video processing of the Berg balance scale parameters [228,229]. In these studies, the assessments of 360-degree turning and the one-leg stance (both part of BBS) are performed using a video camera, without any additional garments. As the authors state, besides increased accuracy, the method could also provide a solution for self-monitoring to patients. Furthermore, another study that used a mobile app based on the Romberg test found it to have 80% sensitivity and 87% specificity in detecting balance disorders [230].

Although wearable devices performing sEMG are a sensitive and accurate way of measuring changes in spasticity determined by neurorehabilitation, they are not a widely available technology. Therefore, in most clinical settings, spasticity is evaluated using standardized tests, such as the Ashworth scale (AS) or the modified Ashworth scale (MAS). Most studies show an improvement in spasticity for patients with stable MS, measured on these scales, especially after electrostimulation, RAGT and BWSTT [149]. However, using H-reflex as a comparison, some authors have suggested MAS and AS are not able to differentiate reflex from non-reflex forms of spasticity [231]. More research is required to address the impact of rehabilitation on spasticity, and novel tests need to be designed for this challenge.

Treatment for dysphagia can be assessed using either the Mann assessment of swallowing ability (MASA) scale [232] or the penetration–aspiration scale (PAS) [233]. The MASA scale comprises 24 items and has 73% sensitivity and 89% specificity for predicting dysphagia [232]. PAS is used to evaluate the functional improvement of deglutition through the fiber optic endoscopic evaluation of swallowing (FEES). Tarameshlu et al. found significant and sustained improvements in both scores for MS patients following a program of oral motor exercises and swallowing compensation techniques in comparison to those who engaged solely in posture reeducation and diet prescription [232]. Other tests recommended by different authors, which could provide useful insights, are the eating assessment tool (EAT-10) and the more specific Dysphagia in multiple sclerosis (DYMUS) questionnaire [174,234].

The findings regarding the therapeutic goals, approaches and assessment scales for the most common symptoms in multiple sclerosis are summarized in Table 1.

## 5. Emerging Techniques and Future Considerations

Technological development opens up new possibilities in the area of neurorehabilitation, for treatment, diagnosis and progress tracking. The mass production of virtual reality devices started in the 1990s and ever since, the technology has gained attention in various fields of work, including healthcare [241]. In medical rehabilitation, virtual reality has brought novel solutions for a variety of afflictions. VR headsets (HMD or head-mounted displays) have been trialed for patients suffering from Parkinson’s disease, in order to improve their gait pattern [242], and for children with cerebral palsy with the purpose of operating motorized wheelchairs [243] and enhancing their spatial awareness [244]. In multiple sclerosis, VR could provide an alternative to traditional rehabilitation programs, through increasing adherence and motivation in patients [235]. Previous studies revealed the ability of VR-based training to enhance gait [245], balance [239] and upper limb mobility and control [246]. Furthermore, MS patients that are dependent on a wheelchair could benefit from this type of technology [247]. In an inpatient setting, VR exercises can also be combined with other neurorehabilitation procedures, such as FES and robot-assisted training [246]. This technology could be especially beneficial for people that are restricted by their location or financial means, or who are reliant on different types of caregivers [248]. Another important aspect that supports the adoption of VR in neurorehabilitation is that the system is able to receive feedback in real time and automatically adapt the intensity to every individual case [249,250]. The clinician can also access the feedback, and is therefore able to track the progress of every patient and change the settings accordingly [236].

Another novel approach to the treatment of multiple sclerosis is represented by robotic exoskeletons [251]. This represents an alternative to BWSTT and RAGT that brings additional benefits, such as offering severely disabled MS patients the option to engage in over-ground walking, therefore enhancing their chances of functional adaptation through neuroplasticity [252,253,254]. Recent literature provides increasing evidence for the efficacy of robotic exoskeletons, with more pronounced results for gait, balance and mobility improvement in MS patients suffering from more advanced forms of the disease [240,255,256].

Cognitive rehabilitation could also benefit from the introduction of new technologies. Computer-assisted cognitive rehabilitation aims to re-train the residual neurological capacity by creating individualized strategies through cognitive models [257]. It targets the improvement of processing speed, language, attention and memory by making use of specific software and multimedia libraries [40,258,259]. The advantages of computer-assisted cognitive rehabilitation reside in its ability to perform cognitive recovery while providing visual and auditory feedback in real time. Furthermore, it increases adherence and motivation through a diversity of immersive scenarios, and it offers the option for patients to engage in it from home [257].

## 6. Conclusions

Multiple sclerosis is a disease with a wide range of symptoms that has seen an increasing prevalence in recent years and requires a multidisciplinary approach. While neurorehabilitation plays a significant part in the management of symptoms and employs a vast number of approaches for achieving this target, there is a continuous need for updates in the most efficacious therapeutic approaches. More research is required to establish better study designs in order to avoid the current biases related to subjectivity and the impossibility of double blinding. There is also a further need to evaluate the validity and reliability of the tests used to assess the status of the disease and the efficacy of the treatment. Moreover, international collaboration could be useful for establishing protocols comprising rigorously tested and approved exercise programs and physiotherapeutic approaches. In this regard, technological innovation could benefit the area of rehabilitation by introducing the more accurate tracking of treatment responses and novel therapeutic solutions.

## Figures and Tables

**Figure 1 jcm-11-07003-f001:**
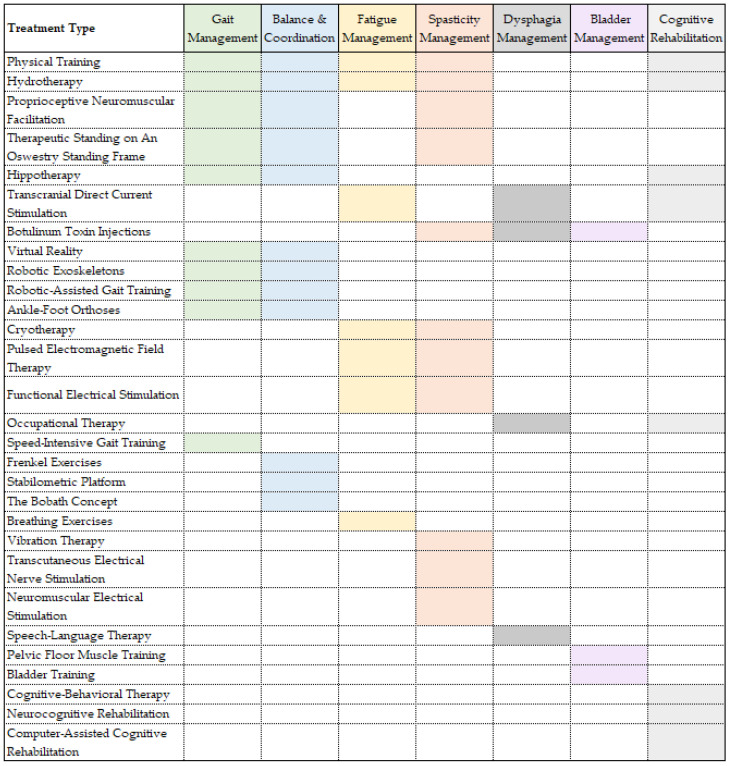
Multiple sclerosis symptoms and rehabilitation treatments. Caption: Rehabilitation treatments ordered by number of symptoms addressed in multiple sclerosis patients.

**Table 1 jcm-11-07003-t001:** Neurorehabilitation for the most common symptoms in multiple sclerosis.

Symptom	Rehabilitation Goals	Method	Assessment Test
**Gait management** (up to 93% of patients after 10 years of diagnosis [91,92])	Increasing lower limb and trunk strength Enhancing gait speed and endurance Improving gait kinematics Maintaining neuroplasticity	Strength training [5] Endurance Training [6] Robotic-assisted gait training [7] Speed-intensive gait training [115] Ankle–foot orthoses [8] Proprioceptive neuromuscular facilitation [130,131] Virtual Reality [235] Robotic Exoskeletons [236]	*Subjective methods:* 2-Minute Walk Test (2MWT) [9] 6-Minute Walk Test (6MWT) [9] Timed 25-Foot Walk test (T25FW) [10] 12-Item Multiple Sclerosis Walking Scale (MSWS-12) [11] Expanded Disability Status Scale (EDSS) *Objective methods:* Wearable sensors combined with surface electromyography (sEMG) [217] Accelerometers [221]
**Balance and coordination management** (80% of cases [237,238])	Preventing falls Enhancing walking stability Posture control Reduce energy requirements Increase continuity of movement	Frenkel exercises [14] Stabilometric platform [15] Hippotherapy [127] The Bobath concept [128] Proprioceptive neuromuscular facilitation [130,131] Virtual Reality [239] Robotic Exoskeletons [236]	*Subjective methods:* Trunk impairment scale (TIS) [222] Berg balance scale (BBS) [223] International cooperative ataxia rating scale (ICARS) [224] *Objective methods:* Video processed BBS [228,229] Mobile apps [230] Nine-hole peg test (NHPT) [226]
**Fatigue management** (75–95% of cases [133,134,135])	Improve mental and physical energy Inflammation reduction Improving depressive symptoms Quality of sleep improvement	Aerobic training [138] Strength exercises [138] Neuromotor exercises (dancing, tai chi, yoga, pilates) [138] Breathing exercises [138] Cryotherapy [141] Pulsed electromagnetic field therapy [143] Functional electrical stimulation [145,146] Hydrotherapy [169]	*Subjective methods:* Quality of Life (QoL) [202,204]
**Spasticity Management** (40–60% of patients [124])	Maintain neuroplasticity Prevent contracture Prevent joint malformation Preserve muscle length Improve ROM of ankle dorsiflexion Decrease hypertonia in the calf muscles Enhance strength of the antigravity muscles	Physical training Vibration therapy Hydrotherapy [168,169] Electrotherapy [158,159] Electromagnetic fields [161,162] Cryotherapy [152,153] Therapeutic standing on an Oswestry standing frame [149] Proprioceptive neuromuscular facilitation [130,131]	*Subjective methods:* Ashworth scale (AS) [149,231] Modified Ashworth Scale (MAS) [149,231] *Objective methods:* Wearable sensors combined with surface electromyography (sEMG) [217]
**Dysphagia management** (around 43% of patients [170])	Speech improvement Avoid malnutrition, dehydration and aspiration pneumonia Maintain healthy weight	Speech–language therapy [173,174] Physical exercises [174] Botulinum toxin injections [174,176,177] Electrotherapy [174] Occupational therapy [174] Transcranial direct current stimulation [178]	*Subjective methods:* Mann assessment of swallowing ability (MASA) [232] Eating assessment tool (EAT-10) [174,234] Dysphagia in multiple sclerosis (DYMUS) [174,234] *Objective methods:* Penetration-aspiration scale (PAS) [233]
**Overactive bladder management** (between 63% and 68% of cases [179])	Increasing resting tension of the pelvic diaphragm Enhanced control over urination mechanism Increase bladder capacity	Pelvic floor muscle training [178,180] Bladder training [182] Weight loss [183] Electrostimulation therapy [74] Botulinum toxin injections [174,180]	*Subjective methods:* Activities of Daily Living (ADL) [16]
**Cognitive Rehabilitation** (34–65% of cases [184])	Reduce emotional disorders Improve emotional control Improve memory, attention and learning Enhance stress management	Cognitive behavioral therapy Neurocognitive rehabilitation [186] Aerobic exercises Transcranial direct current stimulation [192] Computer-assisted cognitive rehabilitation [240]	*Subjective methods:* Quality of Life (QoL) [201,203] Activities of Daily Living (ADL) [208] *Objective methods:* Montreal Cognitive Assessment Test (MoCA)

Neurorehabilitation goals, approaches and assessment tests for the management of the most common symptoms in multiple sclerosis.

## Data Availability

Not applicable.

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
