# Peer review of "Neurorehabilitation in Multiple Sclerosis—A Review of Present Approaches and Future Considerations"

_jcm, 2022, doi:10.3390/jcm11237003_

Round 1

Reviewer 1 Report

The article presents the narrative of a very important topic about the increasingly prevalent disease, MS. However, it needs corrections and additions.

1. Several abbreviations in the paper may need to be clarified for the reader at some stage (some are translated twice - QoL, MRI abbreviation is not developed - line 67, RNS - line 114). It may be worth adding a list of abbreviations at the end of the article.

2. The purpose of the article needs to be clearly defined.

3. The authors write about popular methods for improving MS patients, including the Bobath technique. Also worth mentioning is the PNF method, one of the most famous patient improvement techniques.

4. Adding a graph/schema showing the techniques used in different MS symptoms would be useful. Consider whether the relationship between the type of disorder and the methods used is worth noting, as techniques may be repeated in different symptoms.

5. I suggest changing the title of Chapter 4, as both subjective methods (tests, surveys) and objective methods (using sensors) are mentioned.

6. To be consistent with the methods presented in Chapter Four, I propose to review the available literature on assisted rehabilitation for the automation of balance assessment using BBS. This is an interesting source of information in the context of the analyses carried out. (ex. 1. Romaniszyn, P., Kawa, J., Stępien, P., & Nawrat-Szołtysik, A. (2020). Video-based time assessment in 360 degrees turn Berg balance test. Computerized medical imaging and graphics, 80, 101689; 2. Kawa, J., Stępień, P., Kapko, W., Niedziela, A., & Derejczyk, J. (2018). Leg movement tracking in automatic video-based one-leg stance evaluation. Computerized Medical Imaging and Graphics, 65, 191-199.; 3. Aldenhoven, C. M., Reimer, L. M., & Jonas, S. (2022). mBalance: Detect Postural Imbalance with Mobile Devices. In dHealth 2022 (pp. 30-38). IOS Press.

7. Measurement methods for the analysis of gait kinematics also include systems using accelerometric signals, which the authors do not mention. Please also consider these solutions, as they are very commonly used.

8. Please remove the dots in Table 1 as they interfere with clarity.

9. It might be worth considering whether to highlight objective/subjective methods in Table 1.

10. Chapter 5 should be part of Chapter 4, as the methods described are also techniques used in MS symptom support, as are other new technologies. VR in balance support and the exoskeleton in gait rehabilitation.

Reviewer 2 Report

It is a comprehensive review of current clinical therapy for MS .However ,  phrase organization should be more clear.

1.The article need comprehensively revised for the redundant words and phrase.

2.As for neurorehabilitaiton and NIBS , rTMS should be mentioned.

3.Your article seemed to separate each symptom for consideration.However , MS sometimes presented with these symptom with different severity in different stage.The rehabilitation goal and strategies should be further elucidated.  
